# An auxin-inducible, GAL4-compatible, gene expression system for *Drosophila*

Colin D McClure[1]*[†‡], Amira Hassan[1†], Gabriel N Aughey[1§], Khushbakht Butt[2], Alicia Estacio-Gómez[1], Aneisha Duggal[1], Chee Ying Sia[1], Annika F Barber[2], Tony D Southall[1]*

[1]Department of Life Sciences, Imperial College London, London, United Kingdom; [2]Waksman Institute and Department of Molecular Biology and Biochemistry, Rutgers, the State University of New Jersey, New Brunswick, United States

**\*For correspondence:**
c.mcclure@qub.ac.uk (CDMcC); t.southall@imperial.ac.uk (TDS)

[†]These authors contributed equally to this work

**Present address:** [‡]Queen's University Belfast, School of Biological Sciences, Belfast, United Kingdom; [§]University College London, Department of Clinical and Experimental Epilepsy, UCL Queen Square Institute of Neurology, London, United Kingdom

**Competing interest:** The authors declare that no competing interests exist.

**Abstract** The ability to control transgene expression, both spatially and temporally, is essential for studying model organisms. In *Drosophila*, spatial control is primarily provided by the GAL4/UAS system, whilst temporal control relies on a temperature-sensitive GAL80 (which inhibits GAL4) and drug-inducible systems. However, these are not ideal. Shifting temperature can impact on many physiological and behavioural traits, and the current drug-inducible systems are either leaky, toxic, incompatible with existing GAL4-driver lines, or do not generate effective levels of expression. Here, we describe the auxin-inducible gene expression system (AGES). AGES relies on the auxin-dependent degradation of a ubiquitously expressed GAL80, and therefore, is compatible with existing GAL4-driver lines. Water-soluble auxin is added to fly food at a low, non-lethal, concentration, which induces expression comparable to uninhibited GAL4 expression. The system works in both larvae and adults, providing a stringent, non-lethal, cost-effective, and convenient method for temporally controlling GAL4 activity in *Drosophila*.

## Editor's evaluation

This work will be of interest to *Drosophila* geneticists in its development of a new tool for temporal control of gene induction by the widely-used bipartite Gal4/UAS system. By transferring protein modules from plants that mediate auxin-dependent protein degradation to control the stability of a Gal4-inhibitor (Gal80), the authors successfully demonstrate the ability to control Gal4 activity in flies through the provision of auxin in the food substrate and provide evidence for the sensitivity, specificity, and non-toxicity of this tool.

## Introduction

The ability to manipulate the expression of specific genes in model organisms has been the cornerstone of genetics research over the last 50 years. It is from such studies that geneticists have been able to elucidate fundamental biological process, such as those underlying neurodegenerative diseases (*Lu and Vogel, 2009*), oncogenic mechanisms (*Villegas, 2019*), and neural circuitry involved in memory formation (*Cognigni et al., 2018*).

The fruit fly, *Drosophila melanogaster*, has been at the forefront of genetics since its cultivation in the lab in the early 1900s, by geneticist *Morgan, 1910*. In the 100 years since, a plethora of genetic tools have been developed for this model organism, enabling geneticists to tightly control and manipulate gene expression, allowing them to study the roles of genes in development, physiology, and behaviour. One of the most prominent developments is the GAL4/UAS system, which has been an incredibly powerful tool for the spatial control of transgenes in *Drosophila* (*Brand and Perrimon,*

*1993*). This system relies on a cloned or endogenous promoter to drive the expression of the yeast transcription factor GAL4, which can then bind to the UAS (upstream activating sequence) sites upstream of a transgene and activate its expression. Since the development of this system, *Drosophila* researchers have created thousands of genetically distinct lines to enable expression of transgenes in specific populations of cells (e.g. *Jenett et al., 2012*; *Robie et al., 2017*). The majority of these lines are publicly available at the Bloomington *Drosophila* Stock Center (BDSC) and Vienna *Drosophila* Resource Center (VDRC).

GAL4 activity can be regulated by GAL80, a protein which antagonises GAL4 by inhibiting its activation domain (*Lue et al., 1987*), and thus the ability of the GAL4 to induce transcript expression (*Lee and Luo, 1999*; *Suster et al., 2004*). The generation of a fly line that ubiquitously expressed a temperature-sensitive version of GAL80 (GAL80$^{ts}$) (*Matsumoto et al., 1978*) was a significant advance for the field, as it allowed temporal control of GAL4-induced transgene expression (*McGuire et al., 2003*). Flies can be kept at temperatures permissive of transcript expression (29°C) or temperatures that inhibit expression (18°C), offering researchers to control not only where, but when transgenes are expressed. The versatility of this system enabled researchers to determine time-specific effects of gene function, and is a system that is widely employed within the *Drosophila* community.

Although the impact of this technology on the field cannot be understated, it is becoming increasingly evident that using a temperature shift to induce gene expression is far from optimal for some experiments, especially in fields such as behaviour, neuropathy, and ageing. Flies, as ectotherms, are highly dependent on their ambient temperature, which affects a wide range of physiological and behavioural traits (*Abram et al., 2017*). These include sleep (*Beckwith and French, 2019*), territorial success (*Zamudio et al., 1995*), lifespan (*McCabe and Partridge, 1997*), development rate (*Danjuma et al., 2014*), immunity (*Hunt et al., 2016*), metabolism (*Schou et al., 2017*), and even their microbiome (*Moghadam et al., 2018*). Therefore, the use of a significant temperature shift to induce gene expression should be avoided in these contexts.

As an alternative approach, a range of drug-inducible systems have been developed, where the drug can be administered to flies in their diet to activate transcript expression (*Barwell et al., 2017*; *Kogenaru and Isalan, 2018*; *Osterwalder et al., 2001*; *Potter et al., 2010*; *Sethi and Wang, 2017*). These systems each have advantages and disadvantages (see *Table 1*) and are not often compatible with available GAL4 lines, thus requiring the creation (and optimisation) of their own drivers. In addition, the systems that are compatible with existing GAL4 lines can affect the fly's physiology or survival, thereby limiting their use, and the certainty of any findings when employed. For example, the GeneSwitch system (*Osterwalder et al., 2001*) is not compatible with existing GAL4 lines and requires the user to generate a new modified GAL4 line. Furthermore, the system shows leaky expression in the absence of the drug (RU-486) (*Poirier et al., 2008*; *Scialo et al., 2016*), and RU-486 can cause behavioural changes (*Li and Stavropoulos, 2016*) and is dangerous to handle for pregnant women as it can cause termination of the pregnancy (*Avrech et al., 1991*). Another drug-inducible system utilised in *Drosophila* is the QF system (*Potter et al., 2010*). Here, QF is a transcriptional activator that can be inhibited by QS (analogous to GAL4 and GAL80). Quinic acid inhibits the repressive

**Table 1.** Main advantages and disadvantages of the most common drug-inducible gene expression systems currently available to the *Drosophila* community.

| Expression system | Reference | Advantages | Disadvantages |
|---|---|---|---|
| GeneSwitch | *Osterwalder et al., 2001* | Non-toxic for adult flies | Not compatible with existing GAL4 lines<br>Leaky expression (*Poirier et al., 2008*; *Scialo et al., 2016*)<br>Larval exposure impacts on adult sleep patterns (*Li and Stavropoulos, 2016*)<br>Drug is unsafe to handle for female researchers<br>Drug is expensive |
| QF System | *Potter et al., 2010* | Non-toxic for flies | Not compatible with existing GAL4 lines |
| Tet-off GAL80 | *Barwell et al., 2017* | GAL4/UAS compatible | Long induction time (>5 days)<br>Requires x2 copies of the GAL80 transgene<br>Affects microbiota |
| TMP INDUCIBLE | *Kogenaru and Isalan, 2018*; *Sethi and Wang, 2017* | GAL4/UAS compatible | Currently not compatible for general use with existing GAL4 lines<br>TMP drug is only dissolvable in DMSO (affects survival of larvae) or has to be added as a dry powder to food |

action of QS, therefore, can switch on transgene expression. Quinic acid is non-toxic for flies, however, the drawback of this system is that it is not compatible with the vast collection of existing GAL4 lines.

The auxin-degron system, first identified in *Arabidopsis*, involves the auxin-dependent ubiquitination, and subsequent degradation, of proteins that are tagged with a specific auxin-inducible degron (AID) sequence (*Dharmasiri et al., 2005*; *Dharmasiri and Estelle, 2002*; *Li et al., 2019*). Its natural function allows for the prompt elimination of Aux/IAA transcription factors in plants and has been adapted for use in other species, most notably *Caenorhabditis elegans*, to artificially target proteins for rapid degradation (*Zhang et al., 2015*). In this system, the F-box protein TIR1, an auxin receptor, binds with conserved proteins Skp1 and Cullin to form a Skp1–Cul1–F-box (SCF) E3 ligase complex which ubiquitinates proteins tagged with an AID sequence in an auxin-dependent manner. The specificity and versatility of the system has made it an effective tool in multiple systems including mice (*Yesbolatova et al., 2020*) and human cell culture (*Li et al., 2019*).

Importantly, the auxin-degron system has been successfully employed in *Drosophila* (*Chen et al., 2018*; *Trost et al., 2016*). Trost and colleagues showed that AID-tagged EYFP and AID-tagged Rux protein levels could be knocked down in S2 cell lines and transgenic *Drosophila*, respectively. While Chen and colleagues knocked down AID-tagged PERIOD protein in the adult brain. Inspired by these studies, we have applied the auxin-degron system to enable both temporal and spatial control of transgene expression when combined with the GAL4/UAS system in *Drosophila*. This was achieved by creating a ubiquitously expressed GAL80 fused with AID tags, such that GAL80 is degraded in the presence of the auxin phytohormone (*Figure 1*). We demonstrate that this system works in both larvae (5 mM auxin) and adult flies (10 mM), which are concentrations that do not affect fly survival, development, or locomotion (larval crawling and adult climbing). This auxin-inducible gene expression system (AGES) is safe to handle, cheap, easy to prepare, provides tuneable and stringent (i.e. not leaky) expression, and most importantly, is compatible with the majority of existing GAL4-driver lines developed by the *Drosophila* community.

## Results

### Generation of an auxin-degradable GAL80 *Drosophila* line

To generate an auxin-inducible system for control of GAL4 activity in *Drosophila*, we designed a transgene that would ubiquitously express two proteins, TIR1 and AID-tagged GAL80. (*Figure 1*). We included shorter AID sequences (minimal degron of IAA17) than previously used in *Drosophila* (*Trost et al., 2016*). Fusion of this minimal AID to luciferase resulted in a fusion protein with short half-life in plants (~10 min) (*Dreher et al., 2006*). We also instead used *Arabidopsis thaliana* TIR1 (AtTIR1), which has two point mutations (improving affinity and auxin sensitivity) that can deplete nuclear and cytoplasmic proteins in *C. elegans* (*Zhang et al., 2015*). The P2A sequence between AtTIR1 and AID-tagged GAL80 encodes a self-cleaving peptide that is known to work efficiently in *Drosophila* cell culture (*Daniels et al., 2014*). The *AtTIR1-T2A-AID-GAL80-AID* sequence was codon optimised, synthesised, and cloned downstream of the $\alpha$Tub84B promoter in *pattB* (*Figure 1* and *Figure 1— figure supplement 1*). The $\alpha$Tub84B (tubulin) promoter drives expression of the bicistronic sequence in all cells, all of the time, and (in the absence of auxin) will inhibit any GAL4 activity. When present, auxin will tether the TIR1 to the AID sequences, triggering the degradation of GAL80 and the release of GAL4 inhibition (*Figure 1*).

### AGES allows induction of GAL4 activity in the adult fat body

We combined the AGES with *c564*-GAL4 to see if we could get inducible expression in fat body. A nuclear localised GFP (nls-GFP) was used as a reporter of GAL4 activity. We first examined adult female flies that were placed on food containing different concentrations of 1-naphthaleneacetic acid potassium salt (K-NAA) auxin, which has shown to be more water soluble than NAA (*Martinez et al., 2020*). A negative control consisting of fly food without auxin (0 mM auxin) was included to investigate whether there was any leaky GAL4 activity in absence of auxin. After 24 hr, GFP fluorescence in the abdomen was assayed (note that there is some naturally occurring autofluorescence in the fly abdomen). At 5 and 10 mM, GFP fluorescence is clearly detectable in the abdomen of live female flies (*Figure 2A*). Based on this assay, 10 mM for 24 hr results in ~60% of the GFP fluorescence levels seen in the positive control (*Figure 2B*). At 1 mM there is no detectable GFP above background fluorescence,

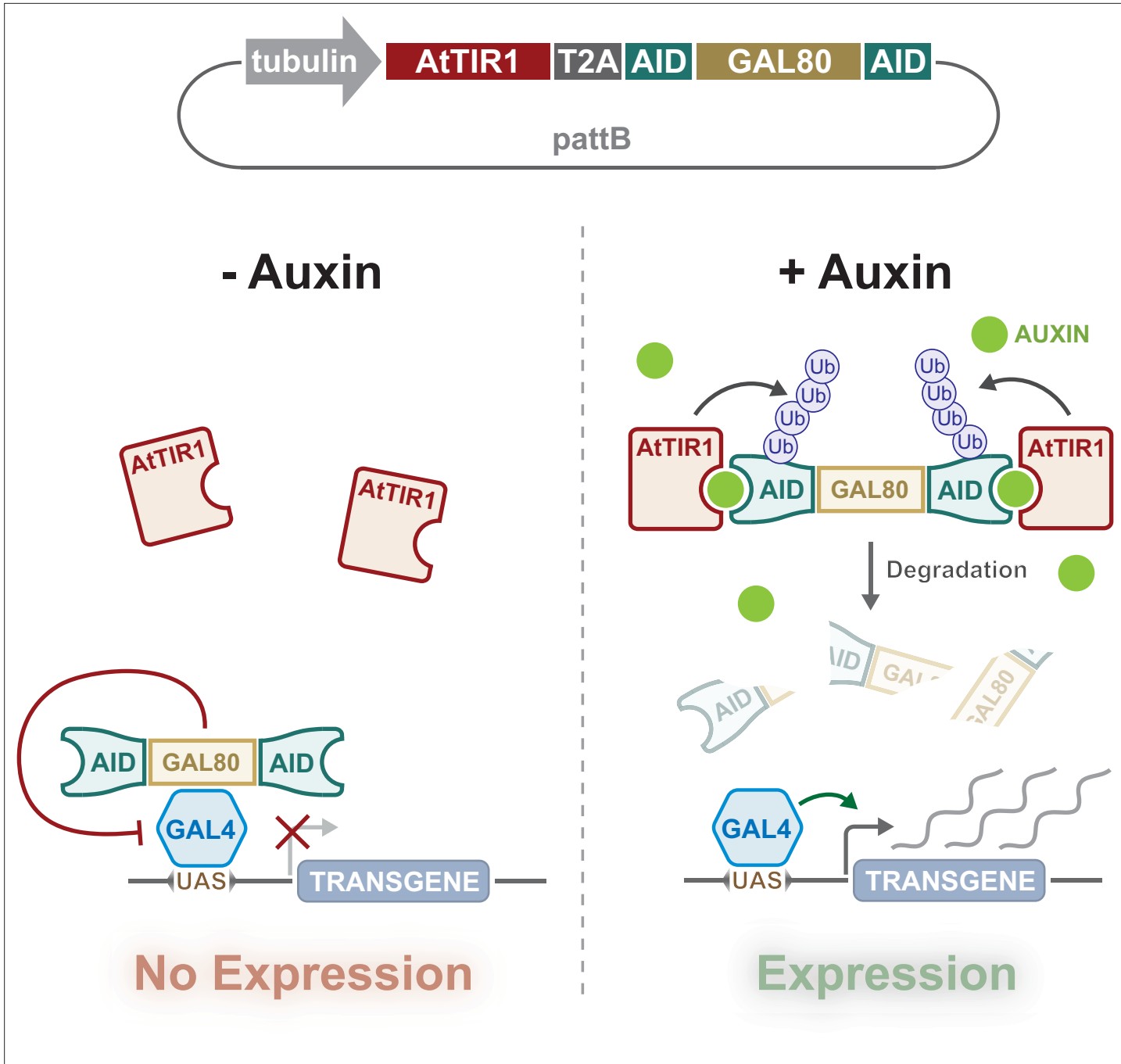

**Figure 1.** Design and action of auxin-inducible gene expression system (AGES). A tubulin promoter ubiquitously expresses the bicistronic TIR1 and AGES cassette. Auxin-inducible degron (AID)-tagged GAL80 is cleaved from the TIR1 due to the presence of the T2A sequence. In the absence of auxin, GAL80 can inhibit GAL4 activity, whilst the presence of auxin induces degradation of GAL80, allowing GAL4 to drive the expression of *UAS*-transgene(s).

The online version of this article includes the following source data and figure supplement(s) for figure 1:

**Figure supplement 1.** Plasmid map of the auxin-inducible gene expression system (AGES) plasmid.

**Figure supplement 1—source data 1.** Genbank (.gb) sequence file for pattB-tubP-AtTIR1-P2A-miniAID-Gal80-miniAID-SV40.

and the levels are indistinguishable from both the 0 mM and the negative control (lacking both the GAL4 driver and the *UAS*-nls-GFP). To provide an alternative and more direct assay of transgene induction, quantitative PCR (qPCR) was used to measure the levels of *GFP* mRNA (*Figure 2C*). Here, 5 and 10 mM show levels of mRNA equivalent to the positive control, while 0 mM showed no significant difference when compared with the negative control.

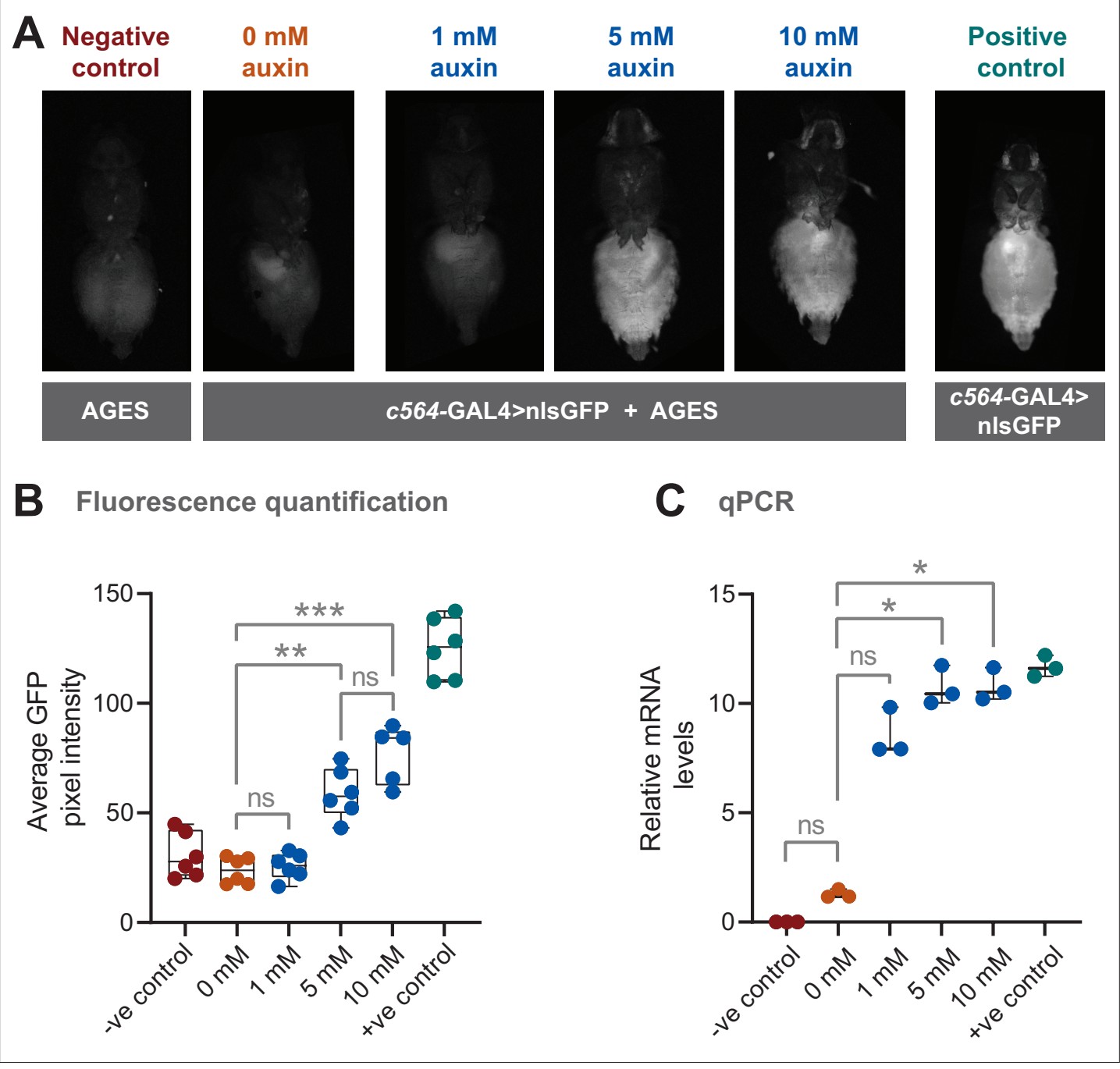

**Figure 2.** Auxin-inducible gene expression system (AGES) effectively induces GAL4 activity in *Drosophila* adults. (**A**) Ventral images of live females that express GAL4 in fat body tissue. Ingestion of food containing auxin (≥5 mM for 24 hr) induces GAL4 activity and the expression of GFP. (**B**) Quantification of GFP levels. Pixel intensity thresholding was performed to isolate abdomens as regions of interest. The average pixel intensities from six replicates were quantified and analysed using Kruskal-Wallis test with Dunn pair-wise comparison (***, $p < 0.001$ and **, $p < 0.01$). (**C**) Quantitative PCR (qPCR) data for *GFP* mRNA levels using different concentrations of auxin (three biological replicates). Values were normalised to housekeeping gene *RpL4* (*Ribosomal Protein L4*) and relative expression levels (compared to the negative control) were calculated using the ΔΔCt method. Y-axis displaying ΔΔCt values and statistics done using Kruskal-Wallis test with Dunn pair-wise comparison (*, $p < 0.05$). See *Figure 2—source data 1* for raw data.

The online version of this article includes the following source data and figure supplement(s) for figure 2:

**Source data 1.** GFP intensity and qPCR values for *Figure 2B, C*.

**Figure supplement 1.** Auxin-inducible gene expression system (AGES) effectively induces GAL4 activity in *Drosophila* adult males.

**Figure supplement 2.** On-off dynamics of auxin-inducible gene expression system (AGES) in adult flies and stability of auxin fly food.

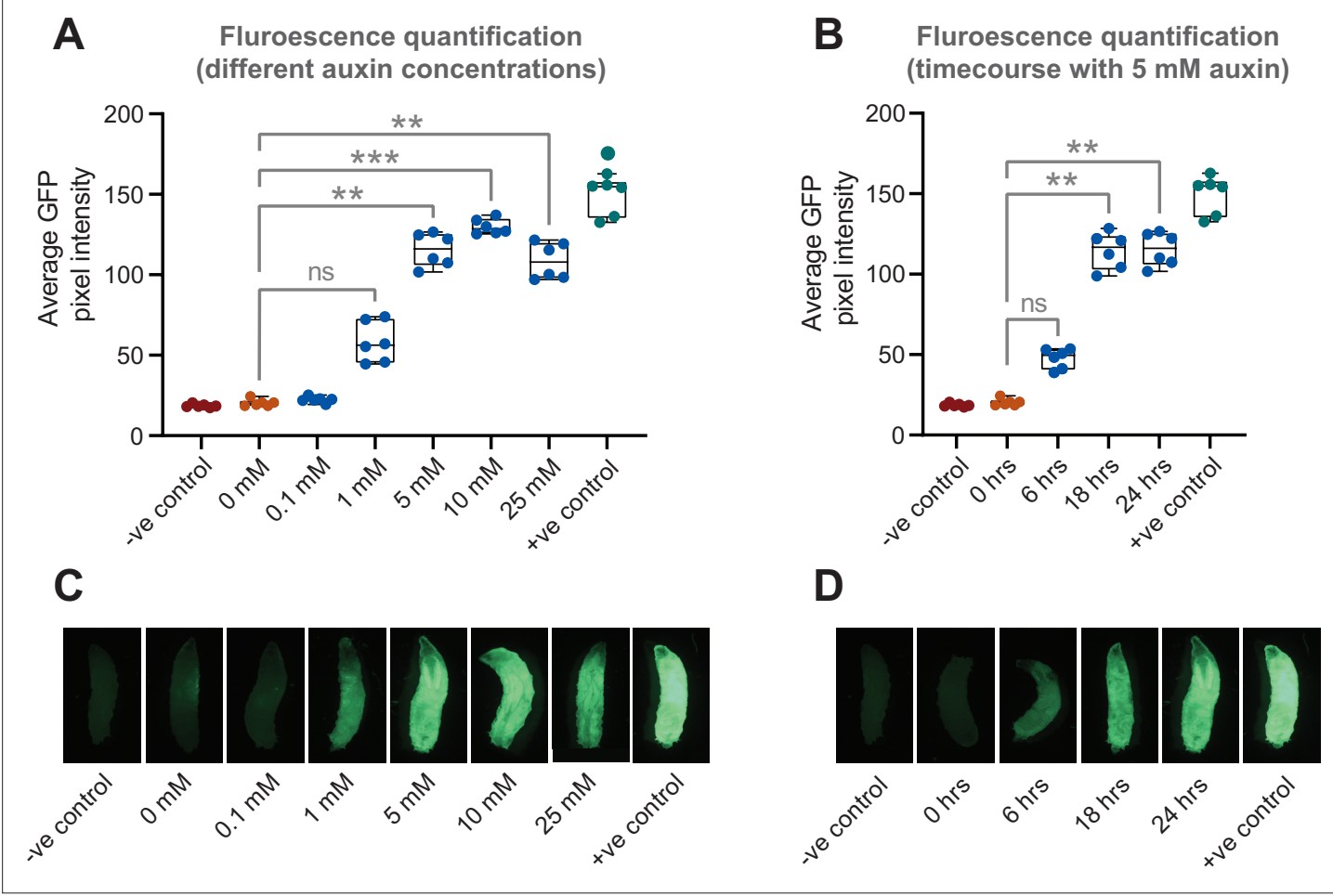

**Figure 3.** Auxin-inducible gene expression system (AGES) allows induction of GAL4 activity in *Drosophila* larvae. (**A**) GFP fluorescence quantification in fat body tissue after induction (for 24 hr) on food with different concentrations of auxin. The average pixel intensities from six larvae were quantified and analysed using Kruskal-Wallis test with Dunn pair-wise comparison (***, p < 0.001 and **, p < 0.01). (**B**) Time course of GFP expression in fat body tissue when using 5 mM auxin. The average pixel intensities from six larvae were quantified and analysed using Kruskal-Wallis test with Dunn pair-wise comparison (**, p < 0.01). (**C**) Representative images of larvae fed on different concentrations of auxin. (**D**) Representative images of larvae imaged at each time interval since induction. See *Figure 3—source data 1* for raw data.

The online version of this article includes the following source data for figure 3:

**Source data 1.** GFP intensity values for *Figure 3*.

Adult males eat less than females (*Wong et al., 2009*), therefore, this will impact on auxin ingestion and possibly on transgene induction. As with females, we examined both GFP fluorescence levels in the abdomen, and GFP mRNA levels in male flies (*Figure 2—figure supplement 1*). Although reduced compared to females, there is still a robust induction of GFP expression.

To assess how quickly transgene expression is turned off after removal of auxin from the food, we performed a time course experiment using adult females and 10 mM auxin. After 24 hr, flies were shifted to food without auxin and GFP fluorescence levels in the abdomen measured every 24 hr. At 48 hr (24 hr without auxin), GFP levels were reduced (*Figure 2—figure supplement 2A*). At 72 and 96 hr, the levels were not significantly different to 0 hr. Furthermore, we tested how stable auxin is in prepared food kept at 4°C. Even after 15 weeks, the auxin food could induce GFP levels to that of newly prepared food (*Figure 2—figure supplement 2B*).

## AGES allows induction of GAL4 activity in *Drosophila* larvae

In addition to the need for temporal control of GAL4 activity in the adult, researchers might also want to induce expression at specific points during larval development. To test if AGES works in larvae,

we used the *c564*-GAL4 fat body driver and tested multiple auxin concentrations (*Figure 3A and C*). Here, we observed that expression could be induced at lower concentrations of auxin. When using food containing no auxin, there is no detectable GFP, whereas with 1 mM auxin, GFP is expressed, although at lower levels than with higher concentrations of auxin (*Figure 3A and C*). Five mM auxin is an optimal concentration, as concentrations above that do not increase GFP levels. Five mM auxin can induce low-level GFP expression after 6 hr (*Figure 3B and D*). However, 18 hr is required to provide maximum levels of expression.

## Use of AGES for controlling cell-specific transgene expression in the larval and adult nervous system

For AGES to work with GAL4 lines expressed in the central nervous system, auxin must be able to pass the selectively-permeable glial membrane, akin to the mammalian blood-brain barrier (*Limmer et al., 2014*). NAA can cross the blood-brain barrier in *Drosophila* (*Chen et al., 2018*). To verify this using K-NAA, we first used *elav*-GAL4 (expressed in all neurons) and examined the expression of GFP in the adult brain. Similar to the fat body *c564*-GAL4 driver, we observe robust expression of GFP, when flies are fed auxin food (10 mM) for 24 hr (*Figure 4—figure supplement 1*). This demonstrates that AGES works in the central nervous system and that K-NAA auxin can cross the blood-brain barrier in *Drosophila*.

We then tested more restricted GAL4-driver lines in the larval ventral nerve cord (*grh*-GAL4) and in the adult (*Or85a*-GAL4). In both cases there was no detectable GAL4 activity in the absence of auxin, while in the presence of auxin (10 mM for adults and 5 mM for larvae) GFP in a pattern consistent with the GAL4 driver is observed (*Figure 4A and C*) at levels significantly above background fluorescence levels (*Figure 4B and D*).

## Effects of auxin on development, adult survival, and behaviour

For a universally applicable drug-inducible system, the drug should not impact on development or mortality of the flies. We tested whether continuous exposure to different concentrations of auxin (K-NAA) would impact on developmental timing. Time to pupation (from egg-laying) is unaffected for concentrations up to 5 mM, however, 10 mM caused a delay of approximately 1 day (*Figure 5A*). Ten mM auxin causes some developmental delay, although it has no impact on the survival of the flies through these developmental stages (compared to a no auxin control) (*Figure 5B*). These data indicate that 10 mM auxin should be avoided for larval induction, however, this is not an issue as >1 mM is sufficient for robust induction of expression (*Figure 5A*) and 5 mM does not cause any developmental delay. Concentrations of 5 and 10 mM auxin can induce expression in adults (*Figure 2* and *Figure 2—figure supplement 1*). Survival assays on both male and female adults show that concentrations of 5 and 10 mM have no effect on survival and lifespan (*Figure 5D and E*). Therefore, in summary, up to 5 mM auxin is optimal for use in larvae, whilst up to 10 mM auxin is best for use in adults. We also tested whether auxin affects locomotion of larvae (crawling assay) and adults (climbing assay) and see no significant effects with 5 and 10 mM auxin, respectively (*Figure 5—figure supplement 1*).

## AGES allows inducible manipulation of *Drosophila* adult circadian locomotor rhythms

To determine whether AGES is suitable for acute adult behavioural manipulation, we recapitulated a classic GeneSwitch manipulation of clock neuron excitability using AGES (*Depetris-Chauvin et al., 2011*). A key circuit for control of *Drosophila* rhythmic behaviour is the ventrolateral neuron (LNv) cluster, of which the small ventrolateral neurons (sLNvs) are both necessary and sufficient for maintenance of free-running locomotor rhythms (*Grima et al., 2004*; *Renn et al., 1999*; *Stoleru et al., 2004*). The four large LNvs and four out of five sLNvs express the neuropeptide pigment dispersing factor (PDF) (*Helfrich-Förster, 1995*; *Renn et al., 1999*). PDF is a key clock output neuropeotide required for synchronisation of clock neuron groups, and loss of PDF function results in arrythmicity, desynchronisation of clock oscillators, and altered period length (*Lear et al., 2009*; *Lin et al., 2004*; *Peng et al., 2003*; *Sheeba et al., 2008*). Adult-specific silencing of *PDF*+ LNvs by expression of the inwardly rectifying potassium channel Kir2.1 using *PDF*-GAL4-GeneSwitch is sufficient to nearly ablate circadian locomotor rhythms in constant conditions without resetting the molecular clock (*Depetris-Chauvin et al., 2011*).

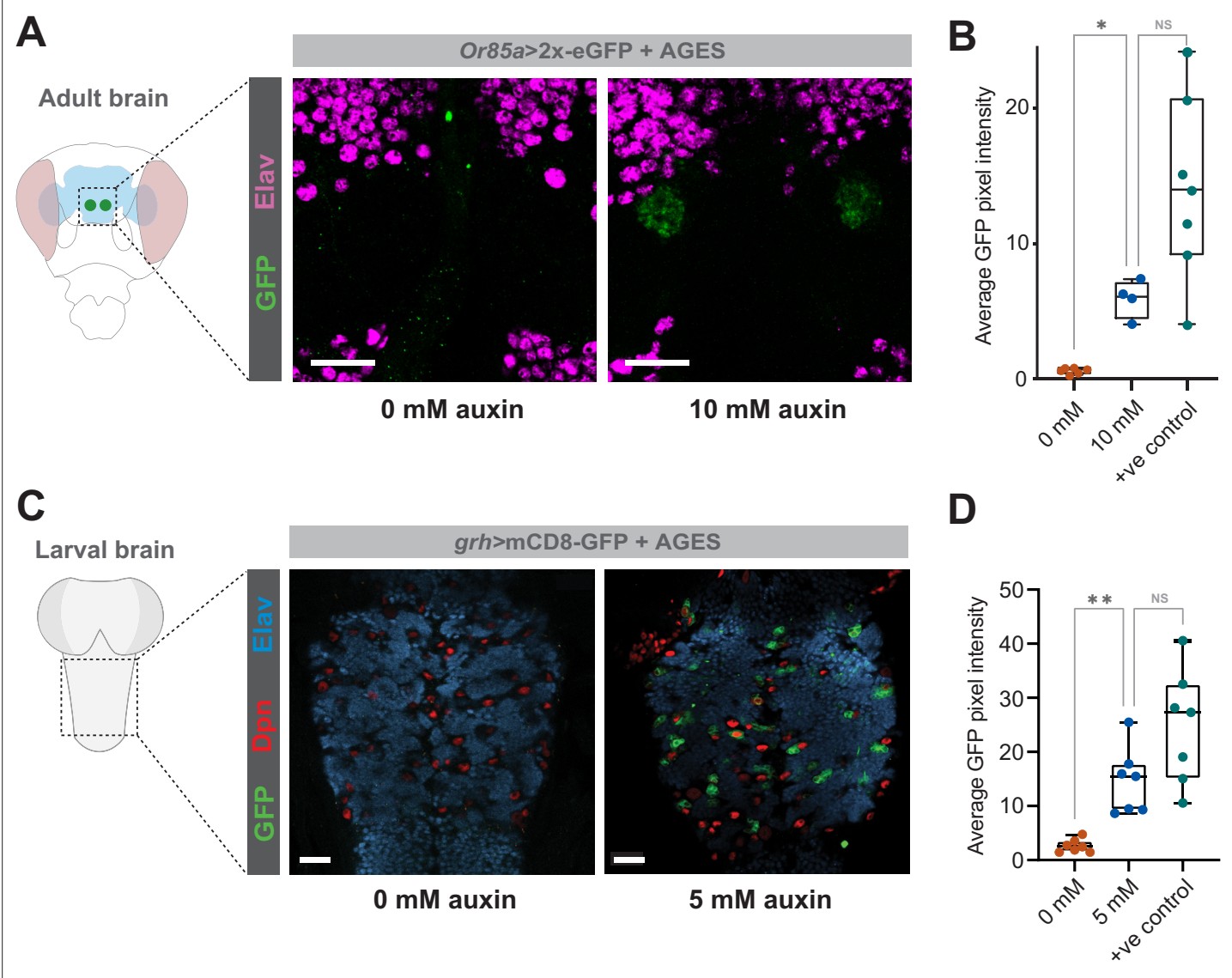

**Figure 4.** Auxin-inducible gene expression system (AGES) allows induction of GAL4 activity in the *Drosophila* adult and larval brain. (**A**) GFP fluorescence driven by *Or85a*-GAL4 in the antennal lobe, with and without auxin. (**B**) GFP fluorescence quantification in the antennal lobe 24 hr after induction (four to six replicates). Statistics performed using ordinary one-way ANOVA (*, p < 0.05). (**C**) GFP fluorescence driven by *grh*-GAL4 in the larval ventral nerve cord, with and without auxin. (**D**) GFP fluorescence quantification in the ventral nerve cord 24 hr after induction (six replicates). Statistics performed using ordinary one-way ANOVA (*, p < 0.05). See *Figure 4—source data 1* for raw data.

The online version of this article includes the following source data and figure supplement(s) for figure 4:

**Source data 1.** GFP intensity values for *Figure 4*.

**Figure supplement 1.** Auxin-inducible gene expression system (AGES) allows induction of pan-neuronal GAL4 activity in the adult brain.

To recapitulate this behavioural experiment, we combined the *PDF*-GAL4 line with the AGES to allow auxin-inducible control of Kir2.1 expression in *PDF+* neurons. We simultaneously replicated GeneSwitch-driven Kir2.1 expression in *PDF+* neurons. After eclosion and entrainment to a 12 hr light:12 hr dark (12:12 LD) schedule on standard fly food, flies were individually loaded into activity tubes and placed in the *Drosophila* activity monitoring (DAM) system (Trikinetics) for 3 days in 12:12 LD, followed by 8 days in constant darkness (DD). Control activity tubes contained standard DAM food (2% agar, 5% sucrose) while experimental food contained RU-486 (200 mg/mL) or NAA (2 or 10 mM). In contrast to the previous experiments, NAA auxin was used here instead of K-NAA auxin. Analysis of the amplitude of the behavioural locomotor rhythm in constant darkness by fast Fourier transform

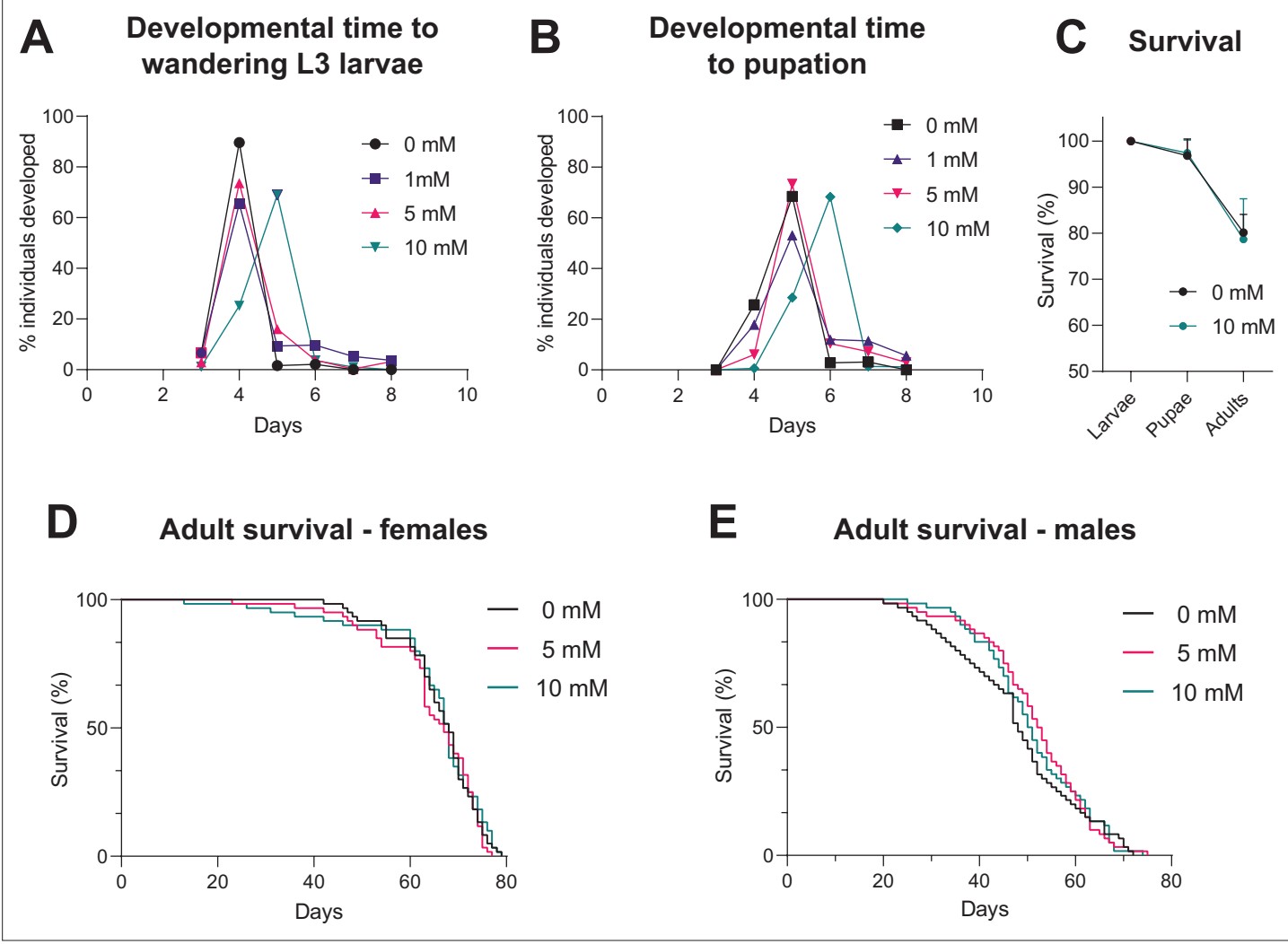

**Figure 5.** Impact of auxin on developmental timing and survival. (**A**) Time taken from egg-laying to wandering L3 larvae. (**B**) Time taken from egg-laying to pupation. (**C**) Survival across developmental stages (larval to pupal and pupal to adult) with 10 mM auxin. (**D**) Survival of adult females during continuous exposure to auxin. (**E**) Survival of adult males during continuous exposure to auxin. Logrank test and weighted Gehan-Breslow-Wilcoxon model (ns) were used for the adult survival assays. See *Figure 5—source data 1* for raw data.

The online version of this article includes the following source data and figure supplement(s) for figure 5:

**Source data 1.** Raw data for larval crawling and adult climbing assays.

**Figure supplement 1.** Working concentrations of auxin do not impact locomotor function of wild-type *Drosophila*.

(FFT) (*Plautz et al., 1997*) showed robust inhibition of locomotor rhythms in both male and female experimental *PDF*-GAL4; AGES > UAS-Kir2.1 flies on 2 or 10 mM NAA (*Figure 6A-D*, *Figure 6—figure supplement 1A,B*). Two mM NAA was sufficient to completely ablate circadian locomotor rhythms in both male and female experimental flies (*Figure 6C and D*), without a significant effect on average 24 hr locomotor activity of parental controls (*Figure 6—figure supplement 2 A-D*). Two mM NAA feeding significantly lengthened the behavioural period of male *PDF*-GAL4/+; AGES/+ parental controls by 19 min, with no effect on females of the same genotype. In agreement with the classic study by *Depetris-Chauvin et al., 2011*, 200 μg/mL RU-486 feeding also reduced circadian locomotor rhythms in both sexes of flies with *PDF*-GAL4-GeneSwitch-driven expression of Kir2.1 (*Figure 6E and F*), though the effect was not as strong as 2 mM NAA feeding in the AGES flies. In addition, RU-486 feeding resulted in a broader distribution of locomotor rhythm strengths and a net increase in rhythmicity in male *PDF*-GAL4-GeneSwitch parental control flies (*Figure 6E*), which was accompanied by a 51 min increased period length and increased average 24 hr locomotor activity (*Figure 6—figure*

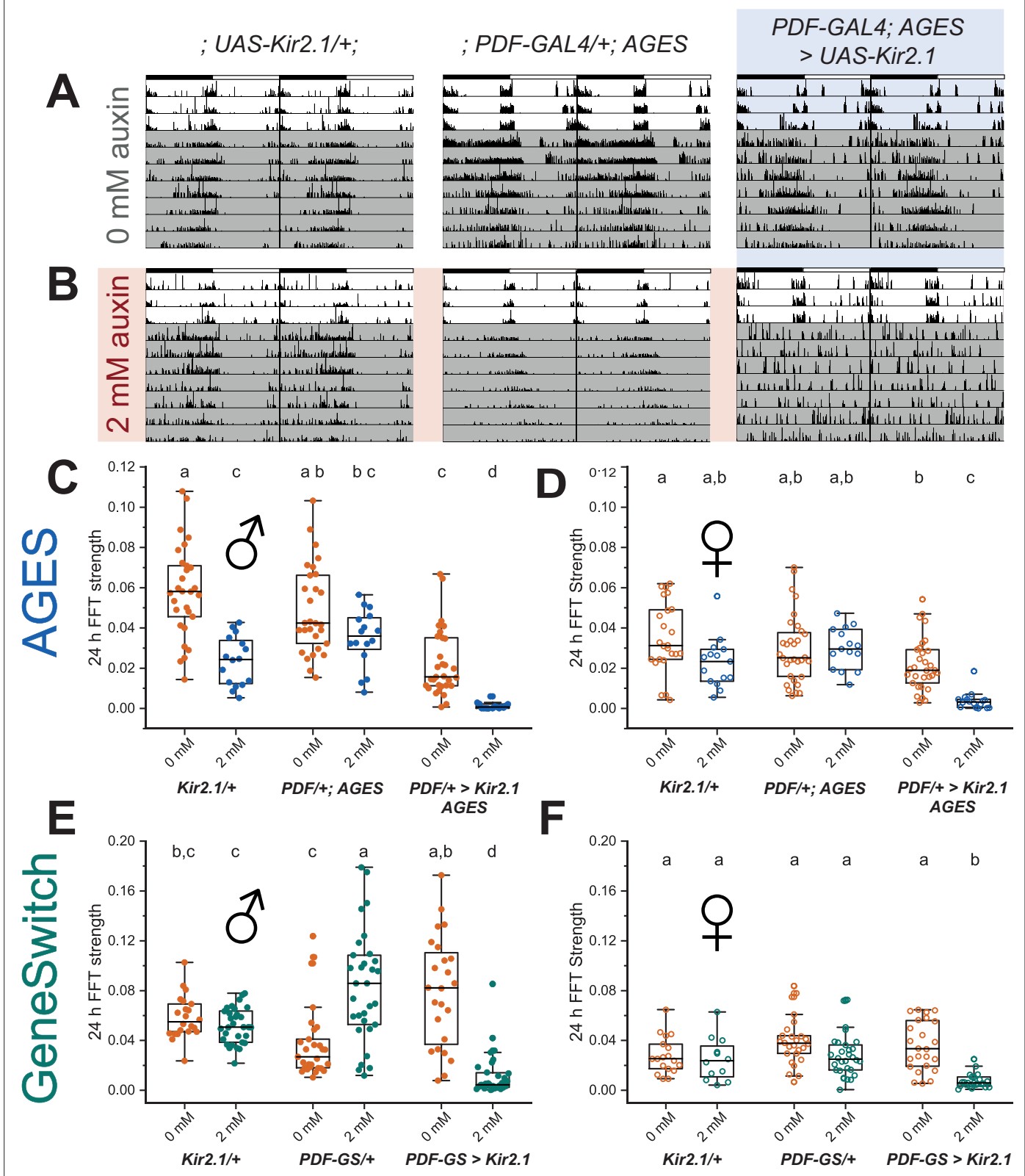

**Figure 6.** Auxin-inducible gene expression system (AGES) induced expression of Kir2.1 in PDF+ clock neurons ablates circadian locomotor rhythms. (**A, B**) Representative double-plotted actograms for 5- to 10-day-old male flies maintained on standard food (**A**) or food supplemented with 2 mm NAA (**B**) for 3 days in 12 hr light:12 hr dark (12:12 LD) and 7 days in DD. Left: parental control *UAS*-Kir2.1, centre: parental control *PDF*-GAL4; AGES, right: experimental *PDF*-GAL4/*UAS*-Kir2.1; AGES/+. Bars indicate LD cycle, grey shaded days indicate constant darkness. (**C**) Amplitude of circadian

*Figure 6 continued on next page*

*Figure 6 continued*

rest:activity rhythms on DD days 2–8 represented by fast Fourier transform (FFT) power at 24 hr for male *PDF*-GAL4;AGES > *UAS*-Kir2.1 flies and their parental controls on standard food (orange) and food supplemented with 2 mM NAA (blue). Points represent individual flies, box shows 25–75% confidence interval, median line, and outliers. (**D**) Twenty-four hr FFT power as in (**C**) for female *PDF*-GAL4;AGES > *UAS*-Kir2.1 flies and their parental controls. (**E**) Twenty-four hr FFT power for as in (**C**) male *PDF*-GAL4-Geneswitch > *UAS*-Kir2.1 flies and their parental controls maintained on vehicle control food (orange) and food supplemented with 466 mM RU-486 (red). (**F**) Twenty-four hr FFT power as in (**E**) for female *PDF*-GAL4-Geneswitch > *UAS*-Kir2.1 flies and their parental controls. For all panels, means were compared by two-way ANOVA by genotype and food substrate, see *Figure 6—source data 1* for raw data, p-values, and key resource data. Means sharing the same letter are not significantly different from one another by Tukey's post hoc test (p > 0.05).

The online version of this article includes the following source data and figure supplement(s) for figure 6:

**Source data 1.** Raw data, p values and key resource data for circadian and activity behavioural experiments.

**Figure supplement 1.** Dose-dependent NAA effects on behaviour of auxin-inducible gene expression system (AGES) parental controls.

**Figure supplement 2.** Effects of auxin-inducible gene expression system (AGES) and GeneSwitch induced expression of Kir2.1 in PDF+ clock neurons on circadian period length and average 24 hr locomotor activity.

supplement 2G). RU-486 also increased the period of female PDF-Gal4-GeneSwitch parental controls by 46 min, with no effect on total activity or rhythm strength (***Figure 6F***, ***Figure 6—figure supplement 2F,H***). In summary, we find that 2 mM NAA feeding is sufficient for AGES-induced expression of Kir2.1 in adult *PDF*+ clock neurons to ablate circadian locomotor behaviour. Indeed, we observed a stronger ablation of locomotor rhythmicity in the AGES compared to GeneSwitch, with fewer off-target behavioural effects of NAA feeding of parental control flies.

## Discussion

The GAL4/UAS/GAL80[ts] systems enable *Drosophila* researchers to control the expression of transgenes both spatially and temporally. However, the impact of the required temperature shift on physiology and behaviour to activate/deactivate the system can affect the experimental results and subsequent conclusions (***Abram et al., 2017***). Drug-inducible systems have been developed as an alternative to the GAL80[ts] method, however, these are not ideal, limited by issues such as leaky expression, long induction times, or incompatibility with existing GAL4 lines. Here, we present AGES, an auxin-inducible gene expression system, which eliminates the need for temperature shifts to manipulate temporal expression and lacks many of the drawbacks of the current drug-inducible systems.

A significant benefit of AGES over existing systems is its compatibility with most existing GAL4-driver lines. Currently, the most prevalent drug-inducible expression systems, that is, the GeneSwitch (***Osterwalder et al., 2001***) and the Q-system (***Potter et al., 2010***), require the generation of their own driver lines. AGES uses a GAL80 fusion protein, which allows the use of the system alongside the majority of existing GAL4-driver lines, whereby suppression of the GAL4-*UAS* activation is achieved in all cells where auxin is absent. Notable exceptions are split-GAL4 lines that do not use the GAL4 activation domain. As with all GAL80-based methods of GAL4 control, the GAL4 line must contain the standard GAL4 activation domain for GAL80 to be affective. There are split-GAL4 lines that use the GAL4 activation domain (***Luan et al., 2006***; ***Pfeiffer et al., 2010***), however, many use a p65 activator domain (***Dionne et al., 2018***; ***Tirian and Dickson, 2017***), or a VP16 activator domain (***Luan et al., 2006***), which will not work with AGES. We have demonstrated the system using multiple driver lines, including *c564*-GAL4, *elav*-GAL4, *grh*-GAL4, *Or85a*-GAL4, and *PDF*-GAL4. These were, in part, used to assess whether the auxin hormone used (K-NAA) could freely access various cell types within the fruit fly, particularly the central nervous system where auxin must pass the selectively permeable glial membrane, akin to the mammalian blood-brain barrier (***Limmer et al., 2014***). Our results, using one fat body GAL4 driver and four different central nervous system GAL4 drivers, demonstrate that auxin can transverse the blood-brain barrier, and that this system should, in theory, be compatible with all other GAL4 lines that contain a GAL4 activation domain.

Successful gene driver systems require negligible expression when un-induced. While some current systems are known to exhibit leaky expression (***Scialo et al., 2016***), AGES demonstrates an almost complete repression of GAL4 activity when un-induced. Using GFP fluorescence as a readout, 0 mM auxin does not show any greater signal than the negative control (***Figures 2B and 4***). Also, the level of *GFP* mRNA without auxin showed no significant difference when compared to the negative control

(*Figure 2C*). The AID tags on GAL80 allow the rapid, and tuneable, degradation of the GAL80 protein, providing users with the option to determine the strength of expression by adjusting the concentration of auxin in the food, or by limiting the exposure time of the flies to auxin.

Temporal control of gene expression is essential for many behavioural experiments in *Drosophila*. We demonstrate that AGES can elicit strong behavioural changes through activating expression of Kir2.1 (an inward-rectifier potassium ion channel) in *PDF*-GAL4 expressing adult neurons (*Figure 6*). Kir2.1 inhibits the activity of PDF neurons resulting in ablation of circadian rhythms. The strong loss of circadian rhythm was observed in both males and females, and even worked at the lower concentration of 2 mM auxin. This is an important development for the *Drosophila* behaviour field, as it allows experiments to be done with existing GAL4 lines, and without the need of a temperature shift or having to use GeneSwitch (*Osterwalder et al., 2001*). While 2 mM auxin food had no effect on the circadian rhythm of female flies (*Figure 6D*) or the *PDF*-GAL4/+ control male flies (*Figure 6C*), it did impact on the *UAS*-Kir2.1/+ control male flies. We currently have no explanation for this. However, it does emphasise the requirement to always perform the appropriate controls. Also, in this context, 10 mM auxin food did have an effect on both male control lines (*Figure 6—figure supplement 1A*). So, while 10 mM auxin had no impact on adult climbing (*Figure 5—figure supplement 1B*), it did impact on circadian rhythms. It should be noted that NAA was used for the circadian rhythm experiment, whereas K-NAA (more water soluble) was used for all the other experiments. Therefore, K-NAA may have less of an effect on the controls as was seen with NAA.

AGES is also a system that is safe for both humans and flies alike. Auxin hormone carries no hazardous concerns for human handling, and no lethal effects were identified for the effective concentrations of 10 mM or below for both adult and larval stages (*Figure 5*). Continuous exposure of larvae to 10 mM auxin does result in some developmental delay; however, it does not affect survival. Higher concentrations of auxin result in increased mortality rates (data not shown) and should be avoided. Users are encouraged to consider this if attempting to optimise exposure conditions for specific experiments, or GAL4-driver lines that have not been assessed in this paper. Overall, based on our data, we recommend the use of up to 5 mM auxin and up to 10 mM for optimal induction in larvae and adults, respectively.

It is also paramount that inducible systems are easy and cheap to apply, to facilitate their practical use in diverse experimental settings. While expenses of the GeneSwitch system average £0.31 (GBP) per vial, AGES costs ~120× less at £0.0025 per vial (authors' estimate). Furthermore, the water-soluble auxin (K-NAA) can be easily added to the food immediately before the food is poured into vials or bottles, requiring no additional safety measures, enabling ease of production. In our experience, auxin-containing food can be stored 4°C for up to 15 weeks (*Figure 2—figure supplement 2B*) where the hormone's potency still persists. Further studies are required to assess the stability and effectiveness of auxin when stored for longer periods of time, or in differing conditions.

The sensitivity of the system is directly related to the amount of auxin present within the tissue, and thus concerns are raised regarding the fly's age and sex, as these factors impact feeding rates (*Carey et al., 2006*; *Wong et al., 2009*). As all experimental adult flies were aged to 5 days post-eclosion, the effects of age on the system are yet to be determined, although lower expression is to be expected in older flies. To address these concerns, alterations to the auxin application could be made, such as a longer exposure (>24 hr), which as we have shown does not impact on survival (*Figure 5D and E*).

Despite the various advantages that AGES offers the fly community, there are still limitations to consider. These include (i) the response time of the system (18–24 hr), (ii) auxin can have subtle effects on adult behaviour (circadian rhythms), and (iii) induced levels of expression are not always as high as the maximum levels obtained without the system. For example, as male adults eat less than their female counterparts, lower expression of the transgene is observed in males (*Figure 2—figure supplement 1*). However, as GAL4-induced expression is inherently strong, reduced transgene expression may not be an issue. This is evident in our circadian rhythm behavioural experiment, where there is a robust response in both males and females when driving expression of *UAS*-Kir2.1 (*Figure 6C and D*). While a 24 hr induction time is not instant, it is both practical and convenient for researchers performing experiments on adult flies in the lab.

There is scope for AGES to be optimised in the future. One approach would be to utilise a mutated version of TIR1 that can accommodate bulky analogues of auxin. Such mutants, in combination with modified auxins (5-Ph-IAA or 5-adamantyl-IAA), allow the degron system to work with much lower

levels of auxin (>500-fold less) (*Yesbolatova et al., 2020*; *Zhang et al., 2022*) and demonstrate a more rapid degradation. Efforts to test these modifications with AGES are underway.

In summary, AGES offers the fly community a cheap, safe, and easy system for temporal transgene expression using existing GAL4 lines. Furthermore, it does not require a shift in temperature changes for induction of gene expression. It is a particularly promising tool for research fields where researchers want to avoid using temperature shifts (e.g. ageing, behavioural genetics, and neuropathology), and will undoubtedly have wide-ranging benefits for multiple fields of study.

# Materials and methods
## Generation of *AGES* line
The *AtTIR1-T2A-AID-GAL80-AID* sequence was synthesised by Twist Biosciences (twistbioscience. com) (*Figure 1—figure supplement 1*). The *GAL80* sequence (from *Saccharomyces cerevisiae*) and the *AtTIR1* sequence (*Zhang et al., 2015*) were codon optimised for *Drosophila* for more effective translation. The *αTub84B* promoter was amplified from a *tubulin-eGFP* plasmid (gift from M Dionne) using the following primers: tub_FWD: GATATCAAGCTTGCACAGGTCC and tub-RV: GTACCTTC ACGCTGTGGATGAGG. The *αTub84B* and *AtTIR1-T2A-AID-GAL80-AID* sequences were cloned into *pattB* (*Bischof et al., 2007*) using Gibson assembly (*Gibson et al., 2009*). Successful clones were sequence verified. Annotated sequence is in *Figure 1—figure supplement 1—source data 1*. The plasmid is publically available at DGRC: the *Drosophila* Genomics Resource Center (DGRC: ): https:// dgrc.bio.indiana.edu/product/View?product=1568 Microinjection was performed by *Cambridge Fly Facility* using the VK00040 line, which has an *attB* site at location att3B (87B10) on chromosomal arm 3R. Injected adult males were collected and mated to $w^{1118}$ virgin females to identify transgenics (orange eyes). *tub*-TIR1-T2A-AID-GAL80-AID is available at the BDSC (stock #92470) or the VDRC Stock Center (stock #311020).

## Fly stocks and food
In this study, *c564*-GAL4, *UAS*-nls-GFP/*CyO-actin*-GAL4-GFP flies were used to perform fluorescent reporter experiments in the adult and larval fat body, and for quantification of *GFP* mRNA expression. Moreover, these flies were used in adult survival assays. *CantonS* flies were used in the developmental survival and developmental timeline experiments. For fluorescent reporter experiments in the adult brain, we used *elav*-GAL4, *UAS*-nls-GFP. For behavioural experiments, *Iso31* flies were used as a background strain (*Ryder et al., 2004*), and we used *PDF-Gal4* (*Park et al., 2000*) and *PDF*-GeneSwitch-GAL4 (BDSC 81116) to drive expression of *UAS*-Kir2.1 (BDSC 6597). Flies were kept at 25°C on and standard *Drosophila* food (recipe in supplementary material) was supplemented (just before being aliquoted into vials/bottles/plates) with auxin (K-NAA available from Phytotech [#N610] or Glentham Life Sciences [GK2088]) at varying concentrations. For behavioural experiments, flies were raised and entrained on standard food, and transferred to activity monitoring tubes containing 2% agar, 5% sucrose, supplemented with indicated concentrations of RU-486 in ethanol (Mifepristone, Sigma-Aldrich) or NAA (Phytotech, #N600).

## Immunohistochemistry
Adult and larval brains were dissected in 1× PBS and fixed for 25 min at room temperature in 4% formaldehyde (methanol free) in 0.3% Triton X-100 PBS (PBST). They were washed four times for 1 hr with 0.3% PBST. Normal goat serum (2% in PBST) was used for tissue blocking (RT, 15 min to 1 hr) and subsequent overnight primary antibody incubation. All tissue washes were done in PBST.

Primary antibodies used were chicken anti-GFP (Abcam #13970, 1:2000), guinea pig anti-Dpn 1:10,000 (*Caygill and Brand, 2017*) and rat anti-Elav 1:500 (Developmental Studies Hybridoma Bank [DSHB]). Secondary antibodies used include Alexa Fluor 488, 545, and 633 at a concentration of 1:200 (Life Technologies) and tissue was incubated for 1.5 hr at room temperature. Tissue was mounted on standard glass slides in Vectashield Mounting medium (Vector Laboratories). Brains were imaged using a Zeiss LSM 510 microscope. Analysis of acquired images was done using Fiji (*Schindelin et al., 2012*).

## Imaging and image analysis
Live imaging of GFP in adult flies and larvae was performed using a Nikon SMZ 1500 microscope. In larvae, to identify earliest auxin induction effects, animals were then placed on 1 mM

auxin-supplemented food and imaged at 0, 1, 5, 10, and 25 hr intervals. Adults were placed on food containing varying auxin concentrations for 24 hr prior to imaging.

Live GFP levels were quantified in six animals per condition (including negative and positive controls). Binary images were created by using a threshold (Outsu's thresholding) and the Wand Tool in Fiji was used to trace the GFP-positive area as region of interest (ROI). The ROI corresponded to abdomens in adults or whole larvae. Mean pixel intensity per ROI was calculated using the Measure plugin in Fiji for a total of six biological replicates. Statistical significance analysed using one-way ANOVA (normally distributed) or Kruskal-Wallis test with Dunn pair-wise comparison (non-normally distributed). Parametric t-test or non-parametric Wilcoxon tests were used to compare individual conditions with respective controls. Statistical tests were completed in *R* (v3.6.3) and plots were performed using the software GraphPad Prism version 9 for Windows.

## Developmental and survival assays

All animals were kept at 25°C. To determine the effects of auxin on larval development, 10 mated Canton-S flies were allowed to lay eggs on fly food containing 0, 1, 5, or 10 mM of auxin for a maximum of 24 hr, or until ~50 eggs were counted. Five replicates were obtained for each concentration, except for 0 mM where four replicates were used. Daily emergence of L3 larvae and presence of pupae were recorded to determine the time required for egg-to-L3, and egg-to-pupae development for each concentration.

Adult survival assays were performed on *c564*-GAL4, *UAS*-nls-GFP/*AGES* flies on three replicates, with 20 flies per replicate. Male and female animals were separated 5 days post-eclosion. Each replicate was continuously exposed to 0, 5, and 10 mM auxin food. Death was scored daily until all flies were deceased. Statistical tests and plots were performed using the software GraphPad Prism version 9 for Windows.

## RNA extraction and real-time qPCR

In order to quantify whether levels of *GFP* mRNA expression changed in the presence of auxin, the *c564*-GAL4, *UAS*-nls-GFP/AGES flies were placed on 0, 5, and 10 mM auxin-supplemented food for 24 hr. The *c564*-GAL4, *UAS*-nls-GFP/+ flies were used as a positive control whilst the *AGES* flies were used as a negative control (lacking both the GAL4 driver and the *UAS*-nls-GFP). The 0 mM was included to investigate whether there was any leaky GAL4 activity in absence of auxin.

Total RNA was extracted from whole adult flies, three per replicate per condition, using a standard TRIzol extraction protocol (*Scientific, 2016*). DNA was degraded using RNAse-free DNAse (Thermo Scientific) and cDNA synthesis was performed using the iScript cDNA synthesis kit (Bio-Rad), following manufacturer's instructions. Real-time qPCR was performed with iTaq Universal SYBR Green Supermix (Bio-Rad) using the StepOnePlus Real-Time PCR System (Applied Biosystems).

The gene *RpL4* (*Ribosomal Protein L4*) was used as reference gene. The *GFP* mRNA expression levels were calculated using the ΔΔCt method (ΔCt = Ct [mean Ct of reference gene] – Ct [target]; ΔΔCt = ΔCt [target] − mean ΔCt [control]) (*Livak and Schmittgen, 2001*). Target refers to the auxin concentrations (0, 5, 10 mM) and control refers to the *AGES* flies. Primers used: RpL4_FW - 5'-TCCA CCTTGAAGAAGGGCTA-3', RpL4_RV – 5'-TTGCGGATCTCCTCAGACTT-3', GFP_FW – 5'-GAGC TGTACAAGAGCAGGCA-3', GFP_RV – 5'-GTTGACGGCGTTTTCGTTCA-3'. Statistical significance analysed Kruskal-Wallis test with Dunn pair-wise comparison in *R* (v3.6.3).

## *Drosophila* activity monitoring assay

We evaluated circadian locomotor rhythms using the DAM system (Trikinetics). Male and female flies of the indicated genotype were entrained in a 12:12 LD cycle prior to loading individual 5- to 7-day-old flies into glass DAM tubes containing control or experimental diets. For AGES flies, the control diet consisted of standard DAM food. For GeneSwitch flies, the control diet consisted of 1% ethanol vehicle in standard DAM food. Flies were monitored via DAM assay for 3 days in 12:12 LD at 25°C, followed by 8 days in constant darkness (DD). Circadian locomotor parameters were analysed using ClockLab software (Actimetrics) for data from days 2 to 8 of DD. Period length was determined by $\chi^2$ periodogram analysis, and relative rhythm power at 24 hr was determined using FFT. FFT power and average daily activity counts for each sex were compared by two-way ANOVA by genotype and food substrate, with Tukey's post hoc comparisons. Because period length could not be estimated

for arrhythmic flies, period length was compared by Student's t-test for flies of the same genotype on different food substrates.

## Acknowledgements

Thanks to Marc Dionne (Imperial College London) for providing the *tubulin-eGFP* plasmid, and to Christian Lehner (University of Zurich) for providing the *pMT-OsTIR1-P2A-H2B-aid-eYFP* plasmid as well invaluable insights into the molecular biology of the auxin-degradation system in *Drosophila*. We thank Andrea Brand for the anti-Dpn antibody. Furthermore, thanks to the entire Southall lab for helpful discussions and pre-reading the manuscript. This work was funded by Wellcome Trust Investigator grant 104567 to TDS, an Innovation Grant from the Society of Developmental Biology to CDM and an NIH-National Institute of Neurological Disorders and Stroke Grant R00 NS105942 to AFB.

## Additional information

### Funding

| Funder | Grant reference number | Author |
| --- | --- | --- |
| Wellcome Trust | 104567/Z/14/Z | Tony D Southall |
| Society of Developmental Biology | | Colin D McClure |
| National Institutes of Health | R00 NS105942 | Annika F Barber |

The funders had no role in study design, data collection and interpretation, or the decision to submit the work for publication.

### Author contributions

Colin D McClure, Conceptualization, Data curation, Funding acquisition, Investigation, Methodology, Supervision, Writing – original draft, Writing – review and editing; Amira Hassan, Data curation, Investigation, Methodology, Visualization, Writing – review and editing; Gabriel N Aughey, Data curation, Investigation, Methodology, Writing – review and editing; Khushbakht Butt, Formal analysis, Investigation; Alicia Estacio-Gómez, Investigation, Validation, Writing – review and editing; Aneisha Duggal, Investigation, Visualization; Chee Ying Sia, Investigation, Writing – review and editing; Annika F Barber, Formal analysis, Investigation, Validation, Writing – review and editing; Tony D Southall, Conceptualization, Funding acquisition, Methodology, Project administration, Supervision, Visualization, Writing – original draft, Writing – review and editing

### Author ORCIDs

Colin D McClure (iD) http://orcid.org/0000-0001-6298-5296
Amira Hassan (iD) http://orcid.org/0000-0003-1640-1602
Tony D Southall (iD) http://orcid.org/0000-0002-8645-4198

### Decision letter and Author response

Decision letter https://doi.org/10.7554/eLife.67598.sa1
Author response https://doi.org/10.7554/eLife.67598.sa2

## Additional files

### Supplementary files
• Transparent reporting form

### Data availability
All data generated or analysed during this study are included in the manuscript and supporting files.

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
