## [Editor Report]

This work will be of interest to *Drosophila* geneticists in its development of a new tool for temporal control of gene induction by the widely-used bipartite Gal4/UAS system. By transferring protein modules from plants that mediate auxin-dependent protein degradation to control the stability of a Gal4-inhibitor (Gal80), the authors successfully demonstrate the ability to control Gal4 activity in flies through the provision of auxin in the food substrate and provide evidence for the sensitivity, specificity, and non-toxicity of this tool.

---

## [Decision Letter]

**Decision letter after peer review:**

Thank you for submitting your article "An auxin-inducible, GAL4-compatible, gene expression system for *Drosophila*" for consideration by *eLife*. Your article has been reviewed by 2 peer reviewers, and the evaluation has been overseen by a Reviewing Editor and K VijayRaghavan as the Senior Editor. The reviewers have opted to remain anonymous.

The reviewers have discussed their reviews with one another, and the Reviewing Editor has drafted this letter to help you prepare a revised submission.

Essential revisions:

It is useful for a new user of this tool to have a more complete description, rather than have to work it out themselves. Currently, the analysis of the tool is modest. More 'proof-of-concept' experiments can be useful and can make the tool more likely to be used. In addition to such experiments, it will be useful if they could have better quantification of the data and provide better controls too.

Please see, from the above perspective, the recommendations of both the reviewers below and the specific suggestions they have made. Could the authors, while embarking on a revised submission, please outline a list of experiments that they plan to do to address the points in the preceding paragraph, including how the current study significantly advances the Trost 2016 work? Given that this study develops on that of the Trost 2016 study, its application toGal80 needs to go further.

*Reviewer #1 (Recommendations for the authors):*

Figure 2B-C: the statistical comparisons being made are not very clear: between the plots at the ends of each bracket? (if so, why not show all comparisons?). In panel C, it is also not clear if the **** refers to comparison of 0 mM with 1 and 5 mM as well as 10 mm.

How do the authors explain the difference between GFP RNA and protein induction? (presumably translation of GFP RNA should not be impacted by the AGES system).

The title is a bit over-hyped: – AGES comprises a modification of Gal80 control; gene expression itself is still dependent upon the Gal4/UAS system.

The number of animals use in different experiments is not clearly indicated.

Line 71: "that are tagged" not "which are tagged"

Line 78: missing word between systems and mice

Figure 2B: panel B, "fluorescence" is misspelt

Figure 2 – figure supp 1: panel B, "fluorescence" is misspelt

*Reviewer #2 (Recommendations for the authors):*

McClure et al. have taken advantage of the Arabidopsis auxin-degron system, which has been shown to function in a wide range of organisms, including *Drosophila*. In this study, the authors cleverly engineer a ubiquitously expressed bicistronic transgene that contains all of the necessary components to inhibit Gal4-induced expression in the absence of auxin. The authors convincingly show their system's efficacy, showing the addition of auxin to food enable Gal4-induced gene expression of a GFP reporter in the fat body, examining both external fluorescence and mRNA levels. GFP expression in the nervous system is a necessary addition due, as the authors state, to the blood-brain barrier. As there was nothing quantitative in Figure 3, a negative control (as used in Figure 2) would have been helpful, as it was hard to determine if 0 mM auxin had no GFP expression visually. As many researchers would be particularly interested in using this system for behavioral studies, further investigation of the system's efficacy for studying behaviors would add to the interest and impact of this work; for example, simple behavioral controls when the flies are reared on auxin, e.g., general locomotion. An excellent addition could be to show that a neuronal effector UAS (for example, csChrimson or TRPA1) does not induce a Gal4-driven behavior in the absence of auxin, and perhaps how long after a shift to auxin behaviors can be induced. Lastly, it is notable that all the shifts in this paper turn Gal4-induced expression from an off to an on state. It would be interesting to investigate the temporal dynamics of the opposite (turning the system from on to off), which would also be of great value to the broader community.

– The paper lacks clarity in parts that may impact its accessibility to a more general audience, for example, the description of the shortcomings of existing systems could be more clearly stated.

– Line 116 "proper induction of GAL4 activity" is quite a vague statement, perhaps the time course (stated as data not shown) could be added to supplemental.

– The authors state in line 173/174 that the AGES system is compatible with the majority of existing split-Gal4 lines, however Gal80 itself is not compatible with most extant split-Gal4 combinations as most activation domain lines are not derived from Gal4.

---

## [Author Response]

Essential revisions:It is useful for a new user of this tool to have a more complete description, rather than have to work it out themselves. Currently, the analysis of the tool is modest. More 'proof-of-concept' experiments can be useful and can make the tool more likely to be used. In addition to such experiments, it will be useful if they could have better quantification of the data and provide better controls too.

We have included several new experiments demonstrating the successful application of AGES. These include switching on GFP expression in a subset of larval brain cells (*grh*GAL4) and a subset of adult brain cells (*Or85a-*GAL4) as shown in our new Figure 4.

Furthermore, we have shown that AGES can be used to elicit changes in adult behaviour (*PDF-*GAL4 driving *UAS-*Kir2.1). Here auxin-supplemented food can disrupt circadian behaviour in both male and female flies (new Figure 6).

Regarding additional quantification and controls, we have:

– Quantified GFP expression levels in larvae on different concentrations of auxin, and over a time course using 5 mM (new Figure 3).

– Quantified on and off temporal dynamics in adult flies using 10 mM auxin (Figure 2 —figure supplement 2A).

– Tested how long auxin food lasts (Figure 2 —figure supplement 2B).

– Confirmed that auxin food does not affect the behaviour of larvae (crawling) or adults (climbing) (Figure 6 —figure supplement 1).

Please see, from the above perspective, the recommendations of both the reviewers below and the specific suggestions they have made. Could the authors, while embarking on a revised submission, please outline a list of experiments that they plan to do to address the points in the preceding paragraph, including how the current study significantly advances the Trost 2016 work? Given that this study develops on that of the Trost 2016 study, its application toGal80 needs to go further.

We agree that we should have introduced/emphasised the Trost paper more in our manuscript. However, we believe that our study advances/differs from the Trost 2016 paper in several important ways:

– We have improved (using the higher affinity AtTIR1 and codon optimised all components) and applied the method for the specific task of temporally controlling GAL4 activity.

– We have demonstrated that this newly engineered system provides very tight control of protein levels (through the readout of GAL4 activity).

– We have investigated and optimised the concentrations of auxin required to activate GAL4 (through the degradation of GAL80) in both larvae and adults.

– We have tested different concentrations of auxin for its effect on survival and development.

– We have shown that it can be used to manipulate adult behaviour.

Reviewer #1 (Recommendations for the authors):Figure 2B-C: the statistical comparisons being made are not very clear: between the plots at the ends of each bracket? (if so, why not show all comparisons?). In panel C, it is also not clear if the **** refers to comparison of 0 mM with 1 and 5 mM as well as 10 mm.

We have included brackets for all the relevant comparisons in the new Figure 2 to make it clearer.

How do the authors explain the difference between GFP RNA and protein induction? (Presumably translation of GFP RNA should not be impacted by the AGES system).

We are not sure why there is a difference between GFP RNA and GFP fluorescence relative to the positive controls. One possibility is that GFP fluorescence is not a linear readout of protein levels.

The title is a bit over-hyped: – AGES comprises a modification of Gal80 control; gene expression itself is still dependent upon the Gal4/UAS system.

We think that the title is still appropriate, and that further clarification would make the title too long.

The number of animals use in different experiments is not clearly indicated.

Thank you, this is now fixed.

Line 71: "that are tagged" not "which are tagged"

Thank you, this is now fixed.

Line 78: missing word between systems and mice

Thank you, this is now fixed.

Figure 2B: panel B, "fluorescence" is misspelt

Thank you, this is now fixed.

Figure 2 – figure supp 1: panel B, "fluorescence" is misspelt

Thank you, this is now fixed.

Reviewer #2 (Recommendations for the authors):McClure et al. have taken advantage of the Arabidopsis auxin-degron system, which has been shown to function in a wide range of organisms, including *Drosophila*. In this study, the authors cleverly engineer a ubiquitously expressed bicistronic transgene that contains all of the necessary components to inhibit Gal4-induced expression in the absence of auxin. The authors convincingly show their system's efficacy, showing the addition of auxin to food enable Gal4-induced gene expression of a GFP reporter in the fat body, examining both external fluorescence and mRNA levels. GFP expression in the nervous system is a necessary addition due, as the authors state, to the blood-brain barrier. As there was nothing quantitative in Figure 3, a negative control (as used in Figure 2) would have been helpful, as it was hard to determine if 0 mM auxin had no GFP expression visually.

We have now included a negative control and quantified GFP expression levels in larvae on different concentrations of auxin, and over a time course using 5 mM (new Figure 3).

As many researchers would be particularly interested in using this system for behavioral studies, further investigation of the system's efficacy for studying behaviors would add to the interest and impact of this work; for example, simple behavioral controls when the flies are reared on auxin, e.g., general locomotion.

We have examined the effect of working concentration (5 mM for larvae and 10 mM for adults) on larval crawling and adult climbing (Figure 6 —figure supplement 1). Here we see no impact of auxin on these behaviours. The circadian experiment data also showed that auxin did not affect locomotor activity (Figure 6 – supplement 2, A-D).

An excellent addition could be to show that a neuronal effector UAS (for example, csChrimson or TRPA1) does not induce a Gal4-driven behavior in the absence of auxin, and perhaps how long after a shift to auxin behaviors can be induced.

We are pleased to show that AGES can be used to elicit changes in adult behaviour (*PDF-*GAL4 driving *UAS-*Kir2.1). Here auxin-supplemented food can disrupt circadian behaviour in both male and female flies (new Figure 6).

Lastly, it is notable that all the shifts in this paper turn Gal4-induced expression from an off to an on state. It would be interesting to investigate the temporal dynamics of the opposite (turning the system from on to off), which would also be of great value to the broader community.

We have quantified on and off temporal dynamics in adult flies using 10 mM auxin (Figure 2 —figure supplement 2A).

– The paper lacks clarity in parts that may impact its accessibility to a more general audience, for example, the description of the shortcomings of existing systems could be more clearly stated.

We have included the descriptions and shortcoming of the two main existing systems (GeneSwitch and QF-QS) in the introduction (lines 66-74).

– Line 116 "proper induction of GAL4 activity" is quite a vague statement, perhaps the time course (stated as data not shown) could be added to supplemental.

Data now in Figure 2 —figure supplement 2A.

– The authors state in line 173/174 that the AGES system is compatible with the majority of existing split-Gal4 lines, however Gal80 itself is not compatible with most extant split-Gal4 combinations as most activation domain lines are not derived from Gal4.

Thank you for pointing this out – we have now emphasised this in the paper (lines 268-273).